# The Impact and Effects of Host Immunogenetics on Infectious Disease Studies Using Non-Human Primates in Biomedical Research

**DOI:** 10.3390/microorganisms12010155

**Published:** 2024-01-12

**Authors:** Neil Berry, Edward T. Mee, Neil Almond, Nicola J. Rose

**Affiliations:** Research & Development—Science, Research and Innovation, Medicines and Healthcare products Regulatory Agency, South Mimms, Hertfordshire EN6 3QG, UK; edward.mee@mhra.gov.uk (E.T.M.); neil.almond@mhra.gov.uk (N.A.); nicola.rose@mhra.gov.uk (N.J.R.)

**Keywords:** non-human primates, host genetics, infectious disease, biomedical research, emerging viruses, global public health

## Abstract

Understanding infectious disease pathogenesis and evaluating novel candidate treatment interventions for human use frequently requires prior or parallel analysis in animal model systems. While rodent species are frequently applied in such studies, there are situations where non-human primate (NHP) species are advantageous or required. These include studies of animals that are anatomically more akin to humans, where there is a need to interrogate the complexity of more advanced biological systems or simply reflect susceptibility to a specific infectious agent. The contribution of different arms of the immune response may be addressed in a variety of NHP species or subspecies in specific physiological compartments. Such studies provide insights into immune repertoires not always possible from human studies. However, genetic variation in outbred NHP models may confound, or significantly impact the outcome of a particular study. Thus, host factors need to be considered when undertaking such studies. Considerable knowledge of the impact of host immunogenetics on infection dynamics was elucidated from HIV/SIV research. NHP models are now important for studies of emerging infections. They have contributed to delineating the pathogenesis of SARS-CoV-2/COVID-19, which identified differences in outcomes attributable to the selected NHP host. Moreover, their use was crucial in evaluating the immunogenicity and efficacy of vaccines against COVID-19 and establishing putative correlates of vaccine protection. More broadly, neglected or highly pathogenic emerging or re-emergent viruses may be studied in selected NHPs. These studies characterise protective immune responses following infection or the administration of candidate immunogens which may be central to the accelerated licensing of new vaccines. Here, we review selected aspects of host immunogenetics, specifically MHC background and TRIM5 polymorphism as exemplars of adaptive and innate immunity, in commonly used Old and New World host species. Understanding this variation within and between NHP species will ensure that this valuable laboratory source is used most effectively to combat established and emerging virus infections and improve human health worldwide.

## 1. Introduction

A number of non-human primate (NHP) species are used in biomedical research. Their use is necessitated to address specific aspects of neurobiology (not addressed in this review) and immunity to infectious diseases not possible in lower animal models due to anatomical or physiological differences or their susceptibility to selected infectious agents. To underpin their use in infectious disease research, some NHP species have been studied in immense detail in relation to their immunogenetics and the role this plays in the outcome of the interplay between the host and pathogen. NHP models have been developed for a wide range of RNA and DNA viruses as previously reviewed [1]. Great apes are protected species whose use in biomedical research is prohibited in most countries and are therefore not considered here, though the comparative major histocompatibility complex (MHC) genetics of great apes have been addressed in the context of humans and NHPs elsewhere by Heijmans et al. [2]. Both Old World Monkey (OWM) and New World Monkey (NWM) species are used in the infectious disease arena, the former featuring in the majority of studies. Among Old World species, Asian macaques (*Macaca* spp.) are most commonly used, hence we will primarily focus on aspects of host genetics among rhesus and cynomolgus macaques. In particular, Indian rhesus macaques represent the most widely used species of OWM NHPs. Cynomolgus macaques were perhaps historically underrepresented in infectious disease research programmes. However, the characterisation of the MHC genetics of Mauritian-origin cynomolgus macaques (MCM) highlighted the distinct advantages of this host, increasing the value of this more readily available subspecies. Among New World hosts, common marmosets (*Callithrix jacchus*) and red-bellied tamarins (*Saguinus labiatus*) have proved effective models for certain types of study, for example, acute hepatitis C infection [3,4] and some emerging viruses such as the Zika virus as covered in Section 9. While less studied than macaques, the aspects of the immunogenetics of these NWM are also considered.

## 2. Importance of Host Immunogenetics in Biomedical Research

Unlike small animals such as mice, rats and rabbits which are typically inbred, NHP species used in biomedical research were historically obtained from wild populations. In more recent decades, NHPs have typically been bred in captivity from founding populations of diverse geographic origins. Whilst the genetic diversity of a breeding colony is limited by that of the founding population and managed husbandry is used to minimise inbreeding for welfare reasons. In the UK and Europe, the experimental use of animals bred from at least an F2 founder population is considered expected good practice such that most captive populations of NHPs are therefore line-bred. We now understand that this line breeding impacts the outcome of experimental studies, since genetic diversity, including at immunogenetic loci, can determine susceptibility to infection. This has been studied most notably in the case of simian immunodeficiency virus (SIV) infection of rhesus macaques, used as a model for HIV. Effective control of SIV infection and longer survival, often in the context of effective anti-viral CD8+ T-cell activity, have been seen where animals have particular MHC genes (e.g., *Mamu*-A*01 and *Mamu*-B*08) [5,6,7,8,9,10,11]. These observations mirror those from numerous clinical studies demonstrating human HLA associations with the control of HIV-1 in distinct, geographically diverse patient cohorts (e.g., *HLA*-B*27 and *HLA*-B*57) [12,13,14,15,16,17,18,19,20]. Identification of such associations between viral control and host genetics have proved extremely valuable in elucidating potential mechanisms of immune control in the SIV/macaque model. However, our understanding remains incomplete.

The identification that host immunogenetic factors alone may determine the outcome of host pathogen interplay adds a further confounding factor to the other practical complexities of undertaking studies in NHP models. This compounds the challenge of designing experiments of suitable statistical power, where group sizes are necessarily limited for ethical and financial reasons. Understanding of the distribution, frequency and functional effects of NHP genes and specific alleles is essential, so that particular genotypes can be included, excluded or evenly distributed across experimental groups of appropriate sizes and assure that study outcomes are determined by known treatment variables.

## 3. Diversity at Non-Human Primate MHC Loci

Humans carry three classical MHC Class I loci (A, B, C), the products of which primarily contribute to antigen presentation and mediating interactions with innate and adaptive immune systems. Many NHP lineages have evolved with an overlapping but distinct set of MHC class I orthologues, some of which have assumed different immune functions from those seen in humans. In fact, the macaque MHC is considerably more polymorphic than the human MHC. Class I loci, for example, have been subject to significant expansion as a result of multiple gene duplications, rearrangements or deletions resulting in rhesus macaque haplotypes containing two or three expressed *Mamu-A* genes and up to 19 distinct *Mamu-B*-like loci. Additionally, complex transcription patterns are observed such that the number of MHC loci, as well as the transcription and expression of MHC products may vary even between two individuals within the same species. This structural polymorphism is in addition to the allelic polymorphism characteristic of most mammalian MHC genes. This complex arrangement posed a significant challenge for early efforts to characterise Class I and Class II MHC components in NHPs. Nevertheless, the rhesus macaque MHC is now more fully characterised as reviewed in more detail elsewhere by Daza-Vamenta et al. [21]. 

While advances in molecular and sequencing technology in the past 20 years have greatly facilitated MHC characterisation at the genomic and transcriptomic levels for both rhesus and cynomolgus macaques [22,23], comprehensive characterisation is inevitably ongoing. Furthermore, transcriptomics technology provides the opportunity for further detailed characterisation of NHP host genomes in terms of MHC gene expression, enabling a more detailed exploration of the influence of immunogenetics on the outcome of pathogen/host interactions. Specific gene microarray technologies permitted early explorations and selected gene expression profiling. This technology is being superseded by whole genome sequencing approaches and cellular RNA-sequencing experimental protocols that can characterise and discriminate detailed interactions at the level of an individual cell within an organism. 

## 4. Geographic Origins and the Genetic Diversity of Asian Macaques

Macaques are widely distributed across Asia. They are considered physiologically, anatomically and genetically closely related to humans and represent the most abundant species used in infectious disease studies. Rhesus macaques (*Macaca mulatta*), cynomolgus macaques (*Macaca fascicularis*) and pig tailed macaques (*Macaca nemestrina*) are the most frequently used species. Despite their gross similarities, key genetic differences within and between species can influence study outcomes. Most notable is MHC haplotype composition, which can reflect the geographical origin of abundantly distributed species. Hence, where preclinical studies are to be undertaken, it is important to establish whether and how such differences may impact the outcome of a study. Furthermore, studies should be carefully designed, and appropriate steps and measures put in place to mitigate their strongest effects. Rhesus macaques (RM) and cynomolgus macaques (CM) typically originate from diverse areas across Asia, including India, China, and parts of the Indonesian archipelago. A non-indigenous population of CM was introduced to the island of Mauritius which has subsequently expanded [24]. 

The genetic characterisation of these macaque populations has revealed considerable variation within and between geographically defined groups, with striking differences in historically isolated populations. Analyses of mitochondrial DNA [25,26,27,28,29,30] and MHC genes [22,31,32,33,34,35] both point to degrees of genetic diversity. Most notably, Mauritian-origin cynomolgus macaques (MCM) are recognised as being less genetically diverse than macaques from other regions, most likely as a result of the small founder population [22,25,28,36,37,38] established on Mauritius around 500 years ago. In other populations, novel MHC polymorphisms have been identified in diverse Southeast Asian macaques [39,40], with Philippine-derived macaques having a lower degree of MHC polymorphism than Vietnamese or Indonesian macaques, although extensive sharing of MHC Class II alleles between rhesus and cynomolgus macaques exists [41].

A clear confounding factor when extrapolating data from NHP models to man, is the degree to which a particular animal species is permissive for the virus of humans. Asian macaques, for example, which are not susceptible to HIV-1, are also not thought to be infected with simian immunodeficiency viruses (SIV) in their natural environment. By contrast, these macaques are susceptible to experimental infection by certain SIV and chimeric SIV/HIV (SHIV) viruses, frequently resulting in an infection and disease profile comparable to that of HIV-1 in humans. In this manner, the macaque model has been well characterised and extensively used not only as a model of HIV-1 pathogenesis and severe immunodeficiency/AIDS in humans, but also in the development and evaluation of vaccines and other therapeutics. However, the macaque species used may lead to different outcomes with inter-species host genetics playing its role. This has been exemplified in studies where the same virus stock has been used in different host species. For example, cynomolgus or Chinese rhesus macaques infected with selected SHIV or SIV strains exhibit an attenuated disease phenotype. Whereas in Indian rhesus macaques, a more pathogenic outcome has been reported [42]. Conversely, a more consistent and persisting infection has been reported in Mauritian cynomolgus macaques challenged with an SIVsm (E660] stock, compared with more variable outcomes observed in Indian rhesus macaques [43]. While it is difficult to elucidate specific genetic determinants for such discrepancies, genetic differences between these host species clearly represent a dominant factor in the outcomes observed. 

## 5. Specific Host MHC Differences in NHPs Used in Biomedical Research

Characterisation of the major differences in host MHCs has been undertaken in most, if not all, of the major NHP species that are commonly used in biomedical research with much focus on Indian rhesus macaque MHC, considered more complex than the human MHC [1,2]. Comparisons and relevance have been made to the human MHC and other gene families [44,45,46,47], with technological advances facilitating progress [48,49,50]. Inroads have also been made into characterization of other subspecies of rhesus macaque, including Burmese and Chinese-origin rhesus macaques [51,52,53]. Genetic studies of other NHP species such as baboons and sooty mangabeys [54,55] have also been described. To facilitate cross-referencing between species, an appropriate nomenclature was developed, summarised and updated [56,57]. Whereby the human MHC is typically referred to as HLA, in NHP species, the genetic locus is referred to by *Mhc* with a suffix comprising an abbreviation of the species name, e.g., *Mamu* (*Macaca mulatta*) in rhesus macaques, *Mafa* (*Macaca fascicularis*) in cynomolgus macaques and *Mane* (*Macaca nemestrina)* in pig-tailed macaques. Typically, following initial use, the *Mhc* prefix is omitted and we have followed this approach. 

In specific infectious disease settings, the role of MHC polymorphism in determining the differential susceptibility of Indian rhesus macaques and cynomolgus macaques to SIV and SHIV infections has been defined. In particular, the role of selected MHC class I and II alleles and CD8+ T cell mediated cellular immune control of virus load is established. Most notably, selected alleles (e.g., *Mamu*-A*-01) have been strongly associated with elite immune controller status in SIV-infected rhesus macaques [5,6,9,10,58,59,60,61] and SHIV infection [62]. Linkages have also been identified between the selected MHC class I haplotype and slow, medium and rapid disease progression following SIV infection. By contrast, it is recognised that some haplotypes may not be reliably predictive of disease outcome [63]. Two class II alleles, *Mamu*-DRB1*100:03 and *Mamu*-DRB1*003:06, have been identified as being enriched among rhesus macaques displaying elite control of SIV infection [8]. The interplay between the acute response to infection, the maturation of responses and the generation and selection of escape variants demonstrate the critical importance of MHC-restricted CD8 T cell control of SIV, delineating a key role for cellular immunity in SIV infection [64,65]. Live attenuated SIV vaccine studies in several SIV/macaque models have deployed different approaches to unravel protective mechanisms, manipulating both the vaccine virus and components of the immune repertoire in macaques, providing insights into the mechanisms of protection from wild-type challenge [66,67,68,69,70]. Although discrepancies exist between early and late protection in different models, a strong effector-memory T cell response, especially in lymphoid tissue, is thought to be beneficial in protecting against a wild-virus challenge. Moreover, NHP studies have further identified novel aspects of anti-viral immunity in rhesus macaques focussing on the role of MHC-E using novel vectors to elicit significant, although not complete, protection from wild-type challenge. Non-canonical CD8+ T cells that recognised a broad spectrum of SIV epitopes which were induced via MHC class II and MHC-E, rather than from a classical CD8+ T cell response, appears beneficial [71]. Further comparisons have delineated this broader relationship between humans and rhesus and cynomolgus macaques [72] whereby the human homologue, (HLA)-E, is now being proposed as a new avenue for HIV vaccine development [73,74].

In recent years, the cynomolgus macaque MHC and its significance has been better characterised, including less well studied populations of CM [75,76,77,78]. HLA and *Mafa* are located on chromosome 6 in humans and chromosome 4 in cynomolgus macaques, classified into sub-regions or classes (Class I-III). Classical class I genes HLA-A, HLA-B are represented by *Mafa* orthologues *Mafa*-A, *Mafa*-B and *Mafa*-I. An orthologue of HLA-C has not been reported in Old World primates. The genomic structure of the *Mafa* region is similar to that of the HLA region, also having undergone the extensive expansion and duplication akin to that seen in the rhesus genome. Eight distinct haplotypes are now recognised in MCMs, designated M1–M8 (nomenclature having been revised from H1–H6). Major alleles expressed from each haplotype have been defined and these eight haplotypes and simple recombinants thereof, represent the vast majority of MHC diversity in MCMs (Figure 1). 

Notably, MHC haplotype H6/M6 is associated with the sustained control of SIVmac251 infection [79] and H3/M3 in the control of a SHIV infection [80] in MCMs. Microsatellite-based genotyping identified that class I MHC H6 (M6) was associated with a reduction in chronic phase viraemia. A less marked but identifiable increase in viraemia was observed in haplotype H5/M5-positive animals. Others identified associations between MHC M3 heterozygotes and the control of SIVmac239 [81]. Together, these studies emphasise that MHC haplotype strongly influences outcome of SIV/SHIV infection, though the precise associations may vary depending on the virus and further refinement of our understanding is required. Mauritian CMs have also been demonstrated to share two common MHC class I alleles that restrict SIV+ CD8 T cells [82,83,84]. Hence, in this context of AIDS-related research genetically defined populations of NHP augments and refines the assessment of key outcomes that could otherwise bias a particular study.

While less is known about cynomolgus macaque MHC class II genes, one exception is the *DQB1* locus. In one study, 33 *DRB*-sequences belonging to 17 allelic lineages were detected in a total of 68 macaques, 58 originating from Mauritius and 10 from China, with the majority of sequences detected in the latter, further confirming the low degree of genetic variation in MCMs. Hence, the *DRB* region in cynomolgus macaques is polymorphic, sharing sequences belonging to the same allelic lineages as in their closest relative, the rhesus macaque. Two exon 2 DNA sequences identified as being identical in both species may represent a trans-species origin. 

The diversity in the MHC, to a large degree, reflects the geographic origin of macaques. The outcome of SIV infections, immune responses and, therefore, the outcome of vaccine studies maybe influenced by the range of diversity in animals included in a study, meaning that a priori knowledge of macaque immunogenetics remains a key factor in effective study design. 

## 6. New World Hosts

The study of New World Monkey genetics has been more limited than Old World Monkeys, partly due to historical usage of apes and OWMs, limited sample availability and the protected status of some species. Considerable differences exist at the genetic level in gene number and polymorphism. Most notably, orthologues of MHC-*A* and -*C* loci have not been identified in NWM species studied to date while MHC-*B* and *-G* loci appear to have undergone expansion following the divergence of the NWMs and higher primates. Mitochondrial DNA studies have identified an evolutionary split in OWMs and NWMs (Catarrhines and Platyrrhines) of ~35 MYA [85]. At the protein level, known MHC class I and class II molecules are broadly similar between NWMs, OWMs and apes, though there may be some differences in their function. An understanding of these differences is key considering what might be the contribution of host immunogenetics on experimental outcomes. 

The classical antigen presentation function performed primarily by MHC-A and -B proteins in humans, apes and OWMs appears to have been assumed by MHC-G in NWMs, representing allelic diversity as assessed in the context of a red-bellied tamarin colony [86]. This reflects analyses of the common marmoset where there is an apparent absence of a classical MHC class I gene repertoire and associated transcription but instead between four and seven different *Caja*-G alleles are present, indicative of an ancestral locus which has undergone expansion [87]. Although such studies of NWM MHC polymorphism have been limited to either small numbers of samples obtained from wild populations [88], or from breeding colonies which may have restricted or skewed representation of the degree of polymorphism [57,89], MHC class I genes in NWMs appear to exhibit less diversity. How this impacts immunological responses and control of infection remains to be fully elucidated. With these caveats in mind, however, a general picture is apparent of more limited genetic diversity within these species and populations studied. A further consideration in any immunogenetic study of NWM is that marmosets and tamarins are known to exhibit chimerism. Females typically give birth to dizygotic twins and stem cells from one may engraft in the other in utero, resulting in genetic and functional chimerism in blood and tissues. The impact of such chimerism on immunity to natural and experimental infections is unknown but adds a further confounding factor to studies using these species. 

## 7. The TRIM5 Protein Family and Innate Immunity in NHP Models

In recent years, the role of innate immunity and antiviral host restriction factors have been brought into sharp focus as a key mechanism for preventing cross-species transmission post virus entry. It is worth noting that host restriction factors, including APOBECs, tetherin and others, first identified in the HIV/primate immunodeficiency virus field are now recognised to represent a significant modulator of virus/host interactions. Thus, variation in the sequence and expression of these genes have potential to impact at multiple stages in the virus lifecycle, including host adaptive immunity and lead to novel areas for potential anti-viral intervention strategies.

The tripartite motif (TRIM) family of proteins, notably TRIM5α, has been of particular interest to the innate immunity field with relevance to NHP models. TRIM5α is responsible for an intracellular block to retroviruses [90,91] and as an innate immune sensor [92]. The basic structure of the TRIM5α locus and protein is depicted in Figure 2: the tripartite motif protein comprises RING, B-box 2, coiled-coil and B30.2 (PRYSPRY) domains. [93]. Macaque TRIM5α is capable of restricting HIV-1 but not SIVmac, a major determinant of potent anti-HIV-1 activity representing a novel innate defence mechanism against retroviruses. TRIM5α appears to have been under positive selection during primate evolution [94,95] with higher non-synonymous/synonymous substitution ratios in PRYSPRY than non-PRYSPRY regions highlighting an adaptive evolutionary role of TRIM5α. 

Further genetic diversity has been identified in some NHP species whereby a cyclophilin (Cyp) A (CypA) domain has been retrotransposed into the TRIM5 locus, resulting in the expression of a TRIMCyp protein (Figure 2) to further confer anti-retroviral activity. Immunogenetic variation in TRIMCyp has been identified in different NHPs. Two lineages of TRIMCyp proteins appear to have arisen independently, in owl monkeys [96,97] and Asian macaques [98,99,100,101,102]. This independent appearance of TRIMCyp chimeras in two primate lineages represents a remarkable example of convergent evolution, whereby a *Macaca* TRIM5Cyp variant appeared 5–10 million years ago in a common ancestor of Asian macaques [100]. This is further supported by identification of a TRIMcyp in *M. nemestrina* [98]. Differences in the geographic origins of NHP species therefore impact TRIM5 variation [103], both the evolutionary effect of TRIM5 polymorphism and TRIMcyp contributing to different anti-retroviral specificities. Hence, multiple alleles in rhesus macaques and sooty mangabeys, with diversity of TRIM5α allele frequencies in Indian and Chinese macaque populations [100,104,105,106,107] displaying divergent anti-retroviral specificity attributed to B30.2 amino acid variation are further suggestive of selective pressures exerted by retroviruses throughout evolution. 

Such diversity at the TRIM5 locus is of interest given the frequency with which different NHP species and subspecies have been used in HIV research, with TRIM5α polymorphism demonstrated to influence SIV susceptibility in rhesus macaques [108,109]. Moreover, TRIM5 polymorphism represents a key host factor in determining cross-species transmission with impacts on virus pressure exerted by the host driving variant emergence [110]. 

It is now recognised that the TRIMCyp isoform occurs in approximately 20% of Indian rhesus macaques [101], but is not observed in Chinese-origin rhesus macaques [100,102,103] with the anti-retroviral activity of a TRIMCyp isoform described in some cynomolgus macaque populations [104]. 

### 7.1. Cyclophilin A and TRIMcyp in Cynolmolgus Macaques

TRIM5α and TRIMCyp allelic diversity between subpopulations of cynomolgus macaques of different geographic origins has been determined [103,111,112,113], with capability to determine SIV susceptibility and the potential to influence experimental study outcomes. For TRIM5α, with a focus on the B30.2/PRYSPRY domain, seven alleles were identified in Indonesian CMs, three of which (designated *Mafa*-9, *Mafa*-11, and *Mafa*-12) represented novel protein sequences for the PRYSPRY region. *Mafa*-4 was identical to a sequence first identified in Indian RM [100]. In Indonesian CMs, all seven TRIM5α alleles appear represented, *Mafa*-4 allele at the highest frequency (40.9% of all chromosomes), *Mafa*-10 presents at 33.0%, with only low frequencies of *Mafa*-8 and *Mafa*-9 [111]. *Mafa*-4 and *Mamu*-4 most likely represents an allele present in a common macaque ancestor conserved during primate evolution. Extended genetic analyses have confirmed differing allelic frequencies with a TRIM5-associated CypA domain sequence occurring in a high proportion (up to 90%) where a retrotransposed Cyp domain allows the formation of a TRIMCyp protein not present in any Mauritian-derived CM to date. 

Moreover, only three TRIM5α alleles have been identified in MCMs: *Mafa*-4 and cynomolgus-specific *Mafa*-8 and *Mafa*-9, but notably no TRIMCyp variants [103,111,112]. These three alleles differ only by three amino acids (M330V and Y389C in *Mafa*-8, and I437V in *Mafa*-9) in the PRYSPRY domain; however, all share the Q339TFP polymorphism, which, in rhesus macaques, is associated with an SIV-permissive phenotype [106,110]. The genotyping of 90 MCMs confirmed the presence of only these three alleles, the *Mafa*-4/4 homozygote constituting 56.7% of the population, with only 4/90 MCMs not carrying the *Mafa*-4 allele [111]. *Mafa*-4 is commonly represented, frequently comprising nearly 75% of all alleles. The lower frequency of TRIM5α alleles and lack of TRIMcyp further reflect the lower genetic diversity in all genetic regions studied to date in MCMs, extending observations based on MHC loci and mitochondrial DNA signatures. 

These factors therefore require consideration in the use of Mauritian CMs for SIV studies though one advantage might be that their limited genetics minimises differences seen in more genetically outbred populations, thus reducing confounding factors in studies designed to elucidate correlates of infection/protection [79,80,81]. Mitochondrial and nuclear DNA polymorphism analyses further suggest the geographic origin of the Mauritian CM to be Sumatra [38] although Java, a neighbouring island, has also been implicated [27]. The prevalence of *Mafa*-4, *Mafa*-8 and *Mafa*-9 in Mauritian CMs therefore likely further reflects the founder effect of this population. The PRYSPRY region demonstrates a high non-synonymous to synonymous amino acid change ratio where 5/13 amino acid changes appear to have arisen through a nucleotide transversion event minimising genetic reversal. This is consistent with observations made from intraspecies comparisons reflecting the high selective pressure on this locus in primate evolution [94,100,105]. Taken together, these findings suggest that genetic diversity across subpopulations of CMs was greater than hitherto recognised.

### 7.2. Impact of TRIM5 Locus Diversity and TRIM5α Expression on Experimental Studies: An SIV Case Study

The relatively high degree of sequence variation in primate TRIM5 gene sequences among rhesus macaque populations has been investigated for impact on outcomes of experimental NHP challenge studies of HIV/AIDS, where TRIM5/TRIMcyp heterogeneity appears to correlate with altered susceptibility to different SIV strains. Specifically, in experimental SIV challenge studies, acute and steady-state viraemia in plasma represents a major readout of virus infection facilitating the interpretation of the efficacy of vaccine and/or anti-viral treatments. Polymorphisms in TRIM5α were correlated with differential viraemic control of wild-type SIV infection, particularly in Rhesus macaques [59,103,108,109]. Hence, specific TRIM5α alleles may need to be specifically excluded from studies or their effects minimised by balancing among test and control groups. 

The characterisation of the impact of TRIM5α on in vivo studies may not be dependent on genotype alone as the TRIM5α gene expression levels may also come into play. Notably, an upregulation of TRIM5α expression was demonstrated in multiple lymphoid tissues of MCMs immediately following vaccination/infection targeted by the virus [114]. However, the restricted range of TRIM5α genotypes and lack of TRIMcyp variants in this species had no or only limited impact on the replication kinetics in vivo of either a SIVmac viral vaccine or a wild-type SIVsmE660 challenge. There was no impact of TRIM5α genotype on outcomes of homologous or heterologous vaccination/challenge studies in this species. The limited spectrum of TRIM5α polymorphism in MCMs appeared to minimize host bias to provide a robust and reproducible model to understand other parameters that may impact on vaccine efficacy. TRIM5α expression induced, in this case, by live-attenuated SIV vaccination irrespective of TRIM5α polymorphism did not appear to play a contributory role in study outcome. Although the transcriptional activity of TRIM5α in lymphoid tissues was upregulated during the acute infection phase, presumably in response to an increase in the level of expressed capsid sequence in individuals susceptible to SIV as infection levels increased over time, neither upregulated transcription or genetic background influenced the outcome of such protection studies conducted with a number of wild-type SIVs in the more genetically conserved MCM host. 

Correlative analyses to identify any association between limited TRIM5α genotypes or expression in Mauritian CMs infected with different SIV strains (either naïve or vaccinated) have so far not proved statistically significant although larger cohort studies may increase this possibility. MHC genotyping remains a high priority, however, given the ability for certain MHC haplotypes to impact of SIV outcomes, particularly those associated with spontaneous control of infection. 

## 8. Transcriptomics Analyses of NHPs

To better understand aspects of human gene expression, in particular, in the context of exposure to infectious agents, the development of high throughput transcriptomics analysis represents a powerful tool. Inevitably, to understand the implications of these types of studies, *a priori* knowledge of the host genetic background and baseline sequence data is a major consideration, where NHP orthologues can be established to interpret human clinical implications. The rhesus macaque genome represents the most widely studied genome in NHP biomedical research. Technological advances have allowed better investigation of reference genome assemblies. This in turn has led to a considerably better understanding of gene content, repeat motifs and organisation and isoform diversity in this species. A synthesis of whole genome sequencing data from over 800 macaques has provided variant analysis in multiple gene clusters and knowledge of SNP frequency and gene organisation. While this inevitably increases the complexity of comparative studies it does provide a rational genetic basis to interpret a wide range of study areas, including those relating to infectious disease [115]. Similar studies have also been reported for cynomolgus macaques. One recent large study using *M. fascicularis* [116] generated an adult cell atlas across 45 tissues using RNAseq/single cell RNAseq to provide superior resolution of this species. Recent genomic insights using whole genome sequencing approaches on the cynomolgus macaque MHC region [50,117] have added to the body of knowledge. Hence, both rhesus and cynomolgus macaque genomes have been described forming an ongoing representation of these two commonly used species in NHP research. Currently the picture is less clear for NWMs. However, the whole genome sequence of the common marmoset representing 2.26-Gb of a female marmoset has permitted genomic comparisons with Old World macaque species [118]. Further, the marmoset transcriptome has been elucidated and reported [119]. These datasets facilitate further biomedical research into gene expression and transcriptional changes, particularly in response to infection in these species. Transcriptomics analyses of NHPs in response to challenge with an infectious agent will further illuminate our understanding of infectious disease processes and host responses.

## 9. NHPs as Model Systems for Emerging Viruses

Emerging viruses represent an ongoing threat to global human health. The SARS-CoV-2/COVID-19 pandemic constituted an unprecedented international infectious disease crisis, sparking an urgent and concerted response from a whole range of scientific disciplines. Animal models, in particular various NHP species, played a crucial role in understanding the broad range of virological and immunological processes that manifested the spectrum of disease pathologies of SARS-CoV-2 during the acute and immediate post-acute infection period. All OWMs studied, including both Indian rhesus and cynomolgus macaques, baboons and African green monkeys were shown to be susceptible to SARS-CoV-2 infection, and thus helped to establish the host range of the virus [120,121,122,123,124,125,126]. Figure 3 summarises the main OWM and NWM species experimentally challenged with SARS-CoV-2. Importantly, the primary receptor for the virus, angiotensin converting enzyme-2 (ACE-2) across all Old World NHPs is 100% conserved at the amino acid level. Nevertheless, the wild-type challenge in immunocompetent hosts resulted in only mild to moderate disease in young, healthy purpose-bred NHPs. NWMs such as marmosets and tamarins appeared less permissive for the virus, displaying limited virus replication and a predominantly asymptomatic phenotype. As a result, NWMs were not widely used as models for the virus [126,127]. 

As part of the rapid response to the pandemic, Old World NHPs played a key role facilitating the development and safety evaluation of effective vaccines against the virus in pre-clinical studies [128,129,130]. A strong humoral response generating neutralising antibodies to the spike (S) protein was highly beneficial in protecting against lung pathology in these hosts following the wild-type challenge. However, the welfare costs/health benefits analysis of using NHPs needs to be carefully evaluated, as alternative small animal models were identified, notably using rodents, which exhibited more significant pathology than that of NHP models. Delineating the range of animal models available for prospective experimental studies in animals of SARS-CoV-2 has detailed both their merits and drawbacks with knowledge gaps identified [131]. 

Detailed aspects of NHP host genetics, while a consideration, represented a more retrospective analysis of NHP studies into SARS-CoV-2. Although the basic principles of host genetics established for other infections were translatable to SARS-CoV-2: e.g., MCMs representing a less diverse genetic study population compared with Indonesian cynomolgus and Indian rhesus macaques which provided more genetically diverse models. Moreover, the urgency of the response needed for a pandemic meant that the selection of NHP models by groups was often driven by local availability. However, the understanding that different NHP models sit across the spectrum of pathogenesis and progression of disease observed in humans following infection with other infectious agents was translated across to this new pathogen. Once established, the spectrum of disease outcomes observed in NHP models were mapped across to the human situation. Moreover, each NHP model provided the ability to tease apart the role of different aspects of host immunity in protection and disease [132,133]. For SARS-CoV-2, both the role of cellular-based immunity involving CD8 T cells and immunoglobulins in immune transfer experiments were evaluated [134,135]. A better understanding of the concerted immunogenicity of COVID-19 vaccines in protecting against the severest effects of lung pathology in NHPs following wild-type virus challenge under controlled conditions contributed to the rapid generation of effective vaccines. This overriding principle will likely also be important in the study of other emerging or re-emergent viruses where studies incorporating transcriptomics and immune repertoire characterisation build on investigations of genetic background undertaken hitherto.

The initial establishment of the host range of a particular infectious agent is therefore crucial. Rhesus and cynomolgus macaques have therefore featured widely in vaccine immunogenicity studies, where multiple components of the host response are brought into play, to facilitate early vaccine development of a wide range of infectious agents. In this regard, emerging but neglected diseases have been characterised using NHP models to good effect using empirical approaches. Examples include Alphaviruses, Filoviruses, or Flaviviruses using different NHPs to provide comparative data across species [1]. When the Zika virus, a hitherto largely neglected mosquito borne Flavivirus, emerged as a major health concern in the Americas, for example, the deployment of animal models represented an important part of the concerted response. Differences in host handling of the virus, when exposed to the same virus inoculum and dose were noted when rhesus and cynomolgus macaques were compared with two New World hosts, tamarins and marmosets [136]. Unlike SARS-CoV-2, Zika replicated to high levels in both New World hosts, whereby all four species represented potential viable models. Hence, although not specifically defined, differences in outcomes will inevitably have a host genetic basis. Understanding novel disease threats and pathogenesis from pathogens of different NHP species [137] or where pathogens are difficult to study, as in the case of viruses inducing highly pathogenic haemorrhagic disease in a given host, disease pathogenesis and putative vaccine efficacy and immunogenicity studies may be undertaken. Studies on Kyasanur forest disease virus and Alkhurma haemorrhagic disease, for example, agents with high pathogenic potential, have been conducted in pig-tailed macaques [138]. Understanding potential correlates of vaccine protection and the most effective immunological responses in immunobridging studies may provide further rational means to progress studies in NHPs and humans. Such complementary approaches to effective vaccine delivery are exemplified for viruses that require high levels of laboratory containment such as Ebola and Marburg viruses [139,140,141], as part of a broad framework of novel experimental approaches to define protective correlates [142] to bridge between experimental data from animal studies in pre-clinical studies through to licensure of effective vaccines for human use [143]. 

Hence, in different settings, defining infection and disease susceptibility in a range of NHP hosts provides an ongoing framework for delineating emerging or recently emerged virus threats which pose a significant or potential threat to human health. Selected and judicial use of NHPs will therefore continue to provide invaluable insights into the host response to novel/emerging threats to more fully elucidate infectious processes. This in turn accelerates development of effective interventions, including vaccines, to combat many of the viruses which pose a significant threat to human health. Elucidating detailed virus/host interactions provides greater understanding of the host in question, in particular immunogenetic responses to a particular infectious agent, enhancing our ability to combat viruses of global health significance. 

## 10. Summary and Conclusions

NHP models have many important roles to play in the development of preventative and therapeutic medicines for infectious disease, whether unravelling the complexities of pathogenesis or evaluating novel intervention strategies in pre-clinical studies. Inevitably, the outcome of pathogen/host interactions will be, in part, determined by the host genetic background. As detailed here for commonly used NHP species, the host MHC can have a significant impact on virus infection outcomes leading to an understanding of the most effective components of protective immunity. Therefore, the selection of a particular NHP species may be driven not only by local or geographical considerations, but also by the anticipated impact a host genetic background may have, whether a more genetically outbred or a more narrow genetic background are to feature in study design. Previous experience or combined knowledge of a particular NHP species in well-studied infection responses may therefore be hugely valuable in translating to a novel infectious agent, as in the case of emerging or urgent global response scenarios, most recently demonstrated in the appearance of pandemic SARS-CoV-2/COVID-19. Even when the effect is anticipated to be relatively modest, host genetics should be borne in mind when designing and interpreting experimental studies, especially where animal numbers are often limiting. The discovery of novel aspects of immunogenetic factors, as in the case of TRIM5 polymorphism in retrovirus restriction as part of the innate immune response, will require regular updating to build these into a more complete understanding of host’s ability to impact on infection processes. Moreover, as scientific information is accrued and new technologies allow us to interrogate legacy samples from previous NHP studies in new ways, scientists will be able to design better studies using NHP’s by controlling host genetic variables that may impact study outcomes. Together these considerations will empower medical researchers to fully utilise the precious resource of NHPs in a wide range of disciplines and improve human health. 

## Figures and Tables

**Figure 1 microorganisms-12-00155-f001:**
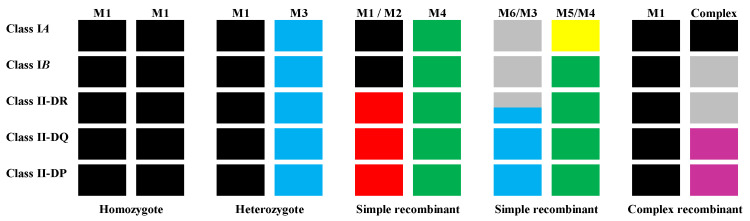
Representation of different haplotypes of Mafa MHCs. Microsatellite data enables the resolution of haplotypes across the ~5 Mbp Mhc region. The majority of haplotypes (~67%) correspond to one of eight intact parental haplotypes, with the remainder being primarily simple recombinants of two parental haplotypes. Complex recombinants of three or more parental haplotypes are observed at low frequency. The major transcribed class I and II alleles have been defined for the parental haplotypes, enabling rapid and low-cost identification of animals with common MHC types.

**Figure 2 microorganisms-12-00155-f002:**
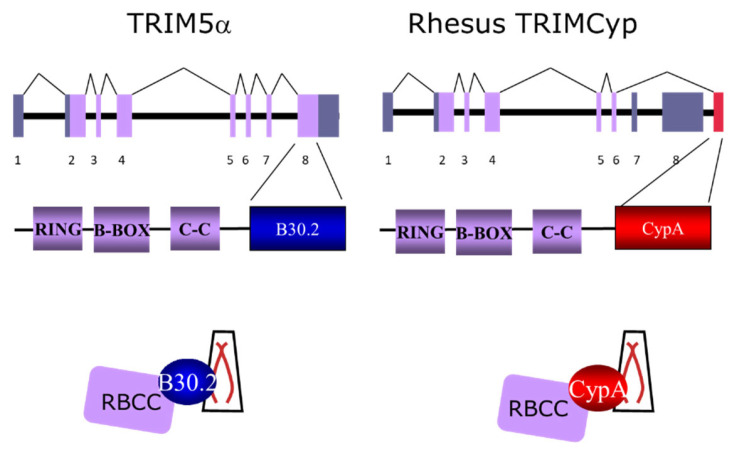
Pictorial representation of interactions between TRIM5α/TRIMcyp components and virus capsid. RBCC, ring B-box coiled coil; B30.2 depicting the PRYSPRY domain which determines retroviral restriction specificity in primates; Cyp A, cyclophilin A. In TRIMcyp the cyclophilin A domain is retrotransposed into the TRIM5 locus. Comparative gene open reading frames for TRIM5α and TRIMcyp are represented 1–8.

**Figure 3 microorganisms-12-00155-f003:**
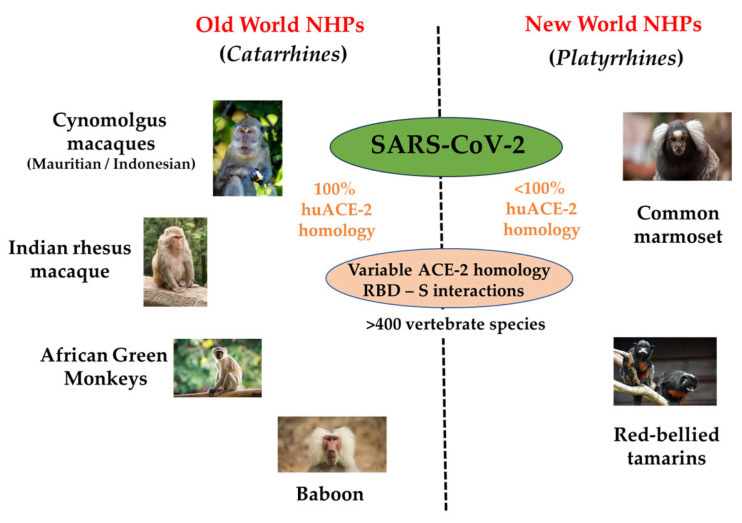
Pictorial representation of the major NHP species used to evaluate different aspects of SARS-CoV-2 species. Outcomes of virus pathogenesis and treatment outcomes, including vaccines, were modelled predominantly in Old World NHPs, where the homology between the primary virus/host receptor (ACE-2) is 100%, though different outcomes exist amongst different species.

## Data Availability

No new data were created or analyzed in this study. Data sharing is not applicable to this article.

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
