# Peer review of "The Impact and Effects of Host Immunogenetics on Infectious Disease Studies Using Non-Human Primates in Biomedical Research"

_microorganisms, 2024, doi:10.3390/microorganisms12010155_

Round 1
Reviewer 1 Report
Comments and Suggestions for Authors
Dear Editor,
At your request, I have reviewed the manuscript entitled “Impact and effects of host immunogenetics in infectious disease studies using non-human primates in biomedical research” by Berry et al., which has been submitted as an review article.
Berry and co-workers wrote a review considering host immunogenetics, including MHC and factors impacting innate immunity, of non-human primate species in the Old and New World monkey lineages and the importance in combating established and emerging virus infections.
To the authors:
The main focus in the first chapters is on the direction of MHC in OWM species. However, the headings of the different chapters are rather general and suggest that genetics in a broader sense will be discussed in the accompanying text. Similarly, for the heading of chapter 7. The heading suggest a broader sense of innate immunity aspects discussed, but only TRIM5 in the context of HIV/SIV is present in the accompanying chapters.
The title of the manuscript also suggest that a broader sense of impact and effects of host immunogenetics in infectious disease studies using non-human primates is discussed.
Parts in the manuscript are disjointed and rather read like a thesis then a review. They need to be overlooked and rewritten. Some illustrations may be added for the first chapters.
Some sentences in the manuscript are rather long and hard to read, and prefer rephrasing for readability.
Heading of chapter 4 may also need revision into something with “geographic” in it, to more reflect what is discussed in the text.
The authors have to read some details on the history of nomenclature of MHC in NHP, discussed in chapter 5, to incorporate in their text. Historically MhcMamu was used, in which Mhc is now mostly omitted. Also, this chapter needs some further attention and overlooked for duplication of text. Moreover, the way it is written now, it looks like HLA-C and Mafa-I are orthologs (line 167-168). There are some recent papers published on cynomolgus MHC that are not discussed. For instance, Campbell et al., 2009; Creager et al., 2011; Karl et al., 2017; Shortreed et al., 2020; de Groot et al., 2022. And some recent genomic insights on the cynomogus macaque MHC region in Karl et al., 2023, genome research, and Hu et al., 2022, Biology Direct. The chapter ends with a conclusion that is not captured by the heading of the chapter.
There are no references to figures in the text.
Also when revising the text, the authors are encouraged to read the text carefully, and be more precise in their written. For instance, in chapter 6, they write “…. But with differences in function”. However, not all NWM MHC class I and II molecules differ in function.
Chapter 6 refers to that several NWM species are known to exhibit chimerism. I thought chimerism is unique to only tamarins and marmosets?
The chapters 7.1 and 7.2 can be fused and compressed, and please look at duplications in the text. PRYSPRY=B30.2, please use more consistency in the text when referring to this. Chapter 7.3 appears currently on page 9 after chapter 8. In this 7.3 part MHC genotyping appears whereas the heading suggest TRIM5x expression discussion.
For chapter 8: the heading needs specification, thus, impact of what type of data is discussed. Moreover this chapter is rather long and a table providing an overview of the results making the text more concise with a focus on the impact of the differences on experimental studies. Thus, rather review the data, and provide a red line of the results, because this is missing.
Detailed feedback:
Chapter 2:
Line 67: suggest to write HLA-B*27 and HLA-B*57
Chapter4:
This section contains some cryptic sentences that are not understandable. For instance, line 109, line 118-119, line 133-135.
Line 115-116: The sentence suggest that CM were introduced on Mauritius in the past 400-500 years. However, they were introduced once, 400-500 years ago, and expanded thereafter.
Line 124: What is SE meaning?
In line 135-136, are the authors referring to HIV pathogenesis, or pathogenesis in general.
Chapter 5:
Define MCM.
Line 165: revise the sentence, it is defined in your text already that HLA is used in humans. Now the sentence has two times humans in it.
Line 169-171: it is not a clear sentence, with two times similar in it.
Line 172: a reference is needed for M1-M8/H1-H6.
Figure 1, in this chapter, is not readable and it is not clear what they authors want to illustrate with this figure. Maybe there is an alternative way for presenting what they want to illustrate.
Line 193: It states one exception is DQB1. And start in the next sentence with DRB?
Line 193-198: The text has no references.
Chapter 6:
Line 216: The context of the subordinate clause “with this allelic …….” it not understood.
Chapter 7:
Line 251-254: It is unclear what the authors want to say with this sentence.
Line 264: Is it correct that “was” should be “has been”.
Chapter 8:
Line 192: used
Line 297: MCM is introduced in an previous chapter, please incorporate consistency in your text.
Chapter 9:
Line 384-386 and 400-401: Cryptic sentences?
Chapter 10:
Line 411-414: ref is missing.
Figure 4 is not descriptive.
Comments on the Quality of English LanguageThe quality of the English of the manuscript is sufficient.
Author Response
Please find the responses to the comments in the attachment.

Reviewer 2 Report
Comments and Suggestions for Authors
The authors present an important review of the impact of NHP species- and population-specific immunogenetic characteristics on the outcome of infectious disease studies.
The lack of somewhat summarized data on the translational potential of the use of NHPs in vaccine immunogenicity studies could be considered a weak point of the manuscript, as the levels of activation of all components of the immune response observed in preclinical studies are key to the selection of promising vaccine candidates. However, this does not undermine the overall value of the review and I recommend its publication in the current form.
Comments on the Quality of English LanguageAlthough generally the manuscript is exceptionally well-written, the structure of some sentences is overly complicated, which reduces the overall readability:
Minor errors:
1. Lines 14-16. The sentence has to be rewritten
2. Line 19. extra "that"
3. Lines 67-69. The sentence has to be rewritten
4. Lines 212-213. The sentence has to be rewritten
5. Line 292. "used"
6. Line 441. extra "binding" and two dots in the end
7. Lines 450-453. The sentence has to be rewritten
Author Response

(The authors gave the same response as above.)

Round 2
Reviewer 1 Report
Comments and Suggestions for Authors
I recommend a thorough editing of the manuscript. I noticed redundancy both within and between sentences. Additionally, the manuscript contains a notable number of lengthy sentences that require the authors’ attention for further improvement before it becomes acceptable for publication.
Author Response
Please see attachment detailing responses.
